# Repetitive Detection of Aromatic Hydrocarbon Contaminants with Bioluminescent Bioreporters Attached on Tapered Optical Fiber Elements

**DOI:** 10.3390/s20113237

**Published:** 2020-06-06

**Authors:** Jakub Zajíc, Steven Ripp, Josef Trögl, Gabriela Kuncová, Marie Pospíšilová

**Affiliations:** 1Faculty of Biomedical Engineering, Czech Technical University in Prague, 27201 Kladno, Czech Republic; pospim14@fbmi.cvut.cz; 2Center for Environmental Biotechnology, University of Tennessee, Knoxville, TN 37996, USA; saripp@utk.edu; 3Department of Cardiology, Regional Hospital Liberec, 46063 Liberec, Czech Republic; 4Faculty of Environment, Jan Evangelista Purkyně University in Ústí nad Labem, 40096 Ústí nad Labem, Czech Republic; josef.trogl@ujep.cz (J.T.); kuncova@icpf.cas.cz (G.K.); 5Institute of Chemical Process Fundamentals of the ASCR, 16502 Prague, Czech Republic

**Keywords:** whole-cell biosensor, bioluminescent bioreporter, optical fiber biosensor, toluene, *Pseudomonas putida* TVA8, *Escherichia coli* 652T7

## Abstract

In this study, we show the repetitive detection of toluene on a tapered optical fiber element (OFE) with an attached layer of *Pseudomonas putida* TVA8 bioluminescent bioreporters. The bioluminescent cell layer was attached on polished quartz modified with (3-aminopropyl)triethoxysilane (APTES). The repeatability of the preparation of the optical probe and its use was demonstrated with five differently shaped OFEs. The intensity of measured bioluminescence was minimally influenced by the OFE shape, possessing transmittances between 1.41% and 5.00%. OFE probes layered with *P. putida* TVA8 were used to monitor liquid toluene over a two-week period. It was demonstrated that OFE probes layered with positively induced *P. putida* TVA8 bioreporters were reliable detectors of toluene. A toluene concentration of 26.5 mg/L was detected after <30 min after immersion of the probe in the toluene solution. Additional experiments also immobilized constitutively bioluminescent cells of *E. coli* 652T7, on OFEs with polyethyleneimine (PEI). These OFEs were repetitively induced with Lauria-Bertani (LB) nutrient medium. Bioluminescence appeared 15 minutes after immersion of the OFE in LB. A change in pH from 7 to 6 resulted in a decrease in bioluminescence that was not restored following additional nutrient inductions at pH 7. The *E. coli* 652T7 OFE probe was therefore sensitive to negative influences but could not be repetitively used.

## 1. Introduction

Primarily controlled chemicals identified in regulations (i.e., United Nations Environment Programme) are selected due to their toxicity in low concentrations, bioaccumulation potential, persistency, carcinogenicity, and repeated assessment in monitoring programs [1,2]. Due to the widespread use of petroleum products and their chemical properties, the most common and stressed water pollutants are benzene, toluene, ethylbenzene and the xylene isomers (BTEX) [3]. It has been shown that these chemicals have adverse health effects, specifically leukemia, cancer, plastic anemia, or bone-marrow disorders in humans, even at low doses [4,5,6]. Solid-phase microextraction techniques and stir-bar sorptive extraction in combination with chromatographic and mass spectrometric analysis have been widely accepted for the analysis of various water pollutants [7,8].

Bioluminescent bioreporters producing light as a response to the presence of specific pollution, such as organic compounds or metals, have been constructed since 1992 [9]. The development and application of whole cell bioreporters has shown great promise in the laboratory and under controlled conditions [10,11,12]. In recent years, smartphones [13] and drones [7] have been used as whole cell biosensor devices with bioluminescent bioreporters. Regardless of many assays, case studies, and constructions of unique biosensor devices such as bioluminescent bioreporter integrated circuits (BBICs) [8], there is no company or institution that has yet established the commercial utilization of bioluminescent bioreporters for the evaluation of pollutant bioavailability. Chemical analysis based on gas chromatography and mass spectroscopy (GC/MS) remains the preferred method, although its combination with bioluminescent bioreporters might reduce the cost and increase the speed of identifying polluted sites, interpreting the hazard of pollutants, and predicting suitability for biodegradation as well as monitoring of biodegradation. A main reason for the limited use of bioluminescent bioreporters is legislative regulation against the environmental release of genetically manipulated cells, e.g., [14]. To better establish the potential utility of bioluminescent bioreporters and their complementary optical biosensor devices, more empirical evidence of field applications and studies of reproducibility and stability, as well as facilitation of analytical protocols and biosensor constructions, are needed.

Continuous monitoring with whole cell biosensors requires repeated inoculation [7] or immobilization [15] of cells on the sensing element. In comparison with other immobilization techniques, the immobilization of bioluminescent bioreporters by attachment on a surface of the sensing element has two advantages: the formation of a layer of cells attached to a surface does not involve any dropping or printing machines, and the cells adhere tightly to the sensing element to minimize loss of the detected bioluminescence signal.

Reporter cells immobilized on the tip of an optical fiber can enable continuous measurements in small volume samples and remote localities. The tiny dimensions of an optical fiber (diameter of fiber end <600 µm) make it possible to immobilize only a few cells on the fiber tip. However, such a small number of cells provides a low light intensity, resulting in low biosensor sensitivity. To increase the signal intensity, the light coupling efficiency can be increased by etching the fiber tip and increasing the number of adhered reporter cells by encapsulation into alginate beads [16]. An increase in bioluminescence signal intensity has also been achieved by the immobilization of reporter cells on the wider end (Ø 1 mm–1 cm) of a tapered optical fiber element [17,18].

In our previous research [19], we demonstrated bioluminescent monitoring of toluene over a period of 135 days by adherence of *P. putida* TVA8 to the chemically modified wider end of a tapered optical fiber element (OFE). This study aims to further elucidate the characteristics of such an OFE-type biosensor. Five different OFEs and two bacterial bioluminescent bioreporter strains were examined. Apart from *P. putida* TVA8, whose bioluminescence is positively induced by toluene [20,21], we also immobilized *Escherichia coli* 652T7, whose bioluminescence is constitutive and decreases in the presence of biotoxicants or other factors that affect cell viability [22]. The bioluminescence of cell layers of *P. putida* TVA8, adhered to several types of OFEs, were induced daily with toluene. Similarly, adhered cell layers of *E. coli* 652T7 were exposed to Luria-Bertani (LB) nutritional media for two weeks. Transmission profiles of all OFEs were calculated with a software script [17] and the results were compared to measured light intensities.

## 2. Materials and Methods

### 2.1. Materials and Solutions

All chemicals used were purchased from Fisher Scientific. Piranha solution contained concentrated H_2_SO_4_ and 30% H_2_O_2_ in a volume ratio of 7:3. LB medium contained tryptone (10 g), yeast extract (5 g) and NaCl (10 g) dissolved in 1 L of distilled water. For solid medium, 17 g of agar was added. Selective LB medium was supplemented with kanamycin (LB_kan_) at a final concentration of 50 mg/L. Phosphate buffer (PB) contained Na_2_HPO_4_·12H_2_0 (23.637 g) and KH_2_PO_4_ (8.98 g) in 1 L of distilled water. The trace element solution contained H_3_BO_3_ (0.062 g) in 1 L of 1M HCl, than CaCl_2_ (2.94 g), ZnSO_4_·7H_2_O (1.44 g), CuSO_4_·5H_2_O (0.39 g), Na_2_MoO_4_·2H_2_O (0.53 g), MnSO_4_·H_2_O (3.5 g), and FeCl_3_·6H_2_O (5.4 g/L) was added in 1 L of distilled water. The mineral salt medium (MSM) consisted of MgSO_4_·7H_2_O (0.2 g), NH_4_NO_3_ (0.2 g), trace element solution (0.1 mL), ferric chloride solution (0.1 mL), PB (100 mL) and distilled water (900 mL). The yeast minimal medium (YMM) consisted of yeast nitrogen base without amino acids (6.7 g/L), synthetic drop-out supplement Y1774 (1.46 g/L), and glucose to the final concentration of 2% (*w*/*v*). For solid medium, 20 g of agar was added.

The toluene induction solution consisted of MSM (19 mL) and toluene-saturated water (1 mL). The final concentration of toluene was 26.5 mg/L, pH 7.2.

The LB induction solution contained 75% of MSM and 25% of LB medium.

The LB, PB and MSM media were autoclaved at 121 °C for 40 min. The trace elements, ferric chloride, and glucose solutions were sterilized by filtration through polytetrafluoroethylene Nalge Nunc Syringe Filters, 0.2 μm from Fischersci.com (Pittsburgh, PA, USA). The YMM was autoclaved at 121 °C for 20 min and was supplemented with sterile glucose to achieve final 2% concentration.

### 2.2. Microorganisms and Their Cultivation

Bioluminescent bioreporter microorganisms were kindly provided by the University of Tennessee. *Pseudomonas putida* TVA8 is a bioluminescent bioreporter harboring a chromosomal *tod-luxCDABE* fusion [23] and produces bioluminescence in the presence of BTEX (benzene, toluene, ethylbenzene and xylene) and trichlorethylene. The constitutively bioluminescent bacterium *Escherichia coli* 652T7 is a *luxCDABE-*based strain that was used by Du et al. (2015) to monitor the biotoxicity of cellulose nanocrystals [22].

*P. putida* TVA8 and *E. coli* 652T7 cells were separately cultivated on LB_kan_ agar for 48 h at 28 °C and then reinoculated to LB_kan_ broth. The cultures were placed in a shaking incubator at 100 rpm and 28 °C and grown overnight to an optical density at 600 nm (OD_600_) of 0.3 ± 0.15 [18,22].

### 2.3. Tapered Optical Fiber Elements (OFEs)

An OFE manufacturing starts with a preform which is heated in a furnace and drawn into an optical fiber. The narrowing part between the dripped part and the fiber was used as an OFE (Appendix A). The resulting shape is determined by preform size, drawing temperature and weight. The OFEs were kindly donated from the Institute of Photonics and Electronics of the Czech Academy of Sciences. The ends of OFEs were polished. Each OFE was characterized by diameters (*D_x_*) measured in 10 mm distances along its length (*L = Z_max_*) (Figure 1 and Figure 2). For the purpose of software calculations (transmittance), the OFE shapes were approximated by a bi-exponential equation (“*Exp2”*) in Matlab software (Table 1). This software was used by Kalabova et al. (2018) [17] to compare the model simulations to the real bioluminescence measured with OFEs.

#### 2.3.1. Preparation of OFEs for Chemical Modification

The OFEs were washed in acetone and rinsed with deionized water. The wider end of the OFE was immersed in Piranha solution at 70 °C for 30 min, washed again in deionized water, and dried at 110 °C for 1 h. After drying, the wider ends of the OFEs were modified with (3-aminopropyl)triethoxysilane (APTES) or polyethyleneimine (PEI).

#### 2.3.2. Surface Modification of OFE with APTES

The wider end of each OFE was immersed in a solution of 5% (v/v) APTES in dry toluene at ambient temperature for 24 h. Afterwards, the OFE was rinsed with toluene and acetone and finally dried at 110 °C for 1 h (modified protocol from [24]).

#### 2.3.3. Surface Modification of OFE with PEI 

The wider end of the OFE was immersed in a 0.2% (w/v) solution of PEI in deionized water for 30 min and then air-dried [25].

### 2.4. Adsorption of Bioreporter Cells on OFE Modified Surfaces

The APTES-modified end of the OFE was fixed vertically in an Erlenmeyer flask containing 150 mL of LB_kan_ medium to which was added 1 mL of an overnight culture of either *P. putida* TVA8 or *E. coli* 652T7. Cells grew and adsorbed on the wider element end in a shaker at 50 rpm and 28 °C for 4 days (i.e., the minimal time needed for *P. putida* TVA8 cells to be adsorbed [19]).

To increase adsorption of *E. coli* 652T7 to the APTES modified OFE surface, FeCl_3_ was added to the LB_kan_ growth medium at a final concentration of 150 µM [26].

Due to inadequate growth of *E. coli* 652T7 on the APTES modified OFE surface, the PEI modified surface was attempted as an alternative. A 20 mL aliquot of an overnight culture of *E. coli* 652T7 in LB_kan_ medium was centrifuged at 3000 g for 5 min. The pellet was resuspended in 20 mL MSM and centrifuged again at 3000 g for 5 min. The pellet was then resuspended in 20 mL of 0.2% PEI in MSM and left in a shaker for 30 min at 100 rpm and 28 °C. Finally, the culture was centrifuged at 3000 g for 5 min and the pellet resuspended in 20 mL of MSM [25]. The wider end of the PEI modified OFE was then immersed in the 20 mL suspension of *E. coli* 652T7 and shaken at 50 rpm for 30 min and 28 °C.

### 2.5. Measurement of Induced Bioluminescence

The thin end of the OFE was attached to a light guiding cable, which was connected to the Oriel 70680 photon multiplier tube. The accelerating voltage of the photon multiplier tube was set to 850 V and the electrical current was manually read from the Oriel 7070 detection system. Experiments were set-up and performed in a light-tight box. The wider end of the OFE with the adsorbed cells was fixed 4 ± 1 mm from the bottom of a 50 mL glass beaker that was then filled with 10 mL of induction solution (toluene for *P. putida* TVA8 or LB/MSM for *E. coli* 652T7), thereby immersing the wider end of the OFE in the induction solution (Figure 3). Reflective aluminum foil was placed underneath the beaker. The current, proportional to the intensity of bioluminescence, was recorded every 30–60 min for 18 h at an ambient temperature of 21 °C. Every 24 h, the wider end of the OFE was gently washed with MSM using a pipette, and then re-immersed into fresh induction solution. Exceptions to these 24 h washings were the 1–3 day pauses for holidays and weekends. On these days, the element remained immersed in the induction solution for up to 72 h. Five different OFEs were tested (Figure 2) with each one being measured over a period of 14–20 days.

### 2.6. Visualization of OFE Cell Adherence Using Scanning Electron Microscopy (SEM)

SEM was used to verify the attachment of *P. putida* TVA8 to its APTES modified OFE. Since the OFEs themselves could not be processed for SEM imaging, quartz cones were used as an alternative (Appendix A). Quartz cones were surface modified with APTES and *P. putida* TVA8 cells were immobilized as explained above. Surface modified quartz cones with adhered cells were placed in a 50 mL beaker containing 30 mL of toluene induction solution. Bioluminescent signaling by the cells was verified by taking light measurements in a Perkin-Elmer IVIS Lumina K imaging system. After two days of immersion, SEM imaging was performed. The quartz cones with immobilized cells were fixed in McDowell–Trump Fixative (Fischer Scientific), gold coated (SPI Module Sputter Coater), and then viewed in a Zeiss Auriga SEM.

### 2.7. Statistics

Five different OFEs were tested (Figure 2). Five OFEs were modified with APES and one was modified with PEI. Each OFE was used once in a single experiment, which lasted 20 days, where light readings were taken every 30–60 min for 18 h. Replication of the immobilization technique on APTES modified quarz surface, and measurement of bioluminescence was shown. Data from the five OFEs plus one OFE from reference [19] were combined. Bioluminescence maxima, peak integrals, and times of the first bioluminescence maxima within the 20 days were plotted and approximated with polynomial and exponential curves respectively.

## 3. Results and Discussion

### 3.1. Imobilization and Induction of Bioluminescence from P. putida TVA8 on APTES Modified OFEs

The APTES modification of OFE led to the successful adherence of *P. putida* TVA8 on its surface. Two days after beginning the immobilization procedure, lumps of cell clusters (100–1000 µm apart) among much smaller scattered clusters or single cells were observed under SEM (Figure 4), which corresponds to standard biofilm establishment characteristics [27]. A fully developed biofilm layer of *P. putida* TVA8 was photographed after 130 days on APTES modified quartz fiber and was presented in a previous paper [19].

The time-records of daily inductions of the five different OFEs are shown in Figure 5. 

The intensities of the detected light were low during the first few days after induction and then gradually increased (Figure 6). This might be ascribed to an advanced covering of the base of the OFE with cells. Most time-record curves show two peaks.

The times of the first bioluminescence maxima decreased from more than 10 h to 2–5 h after the fifth day of induction. The same times of the first maxima were observed in a previous study (Figure 7) [19]. The second bioluminescence maxima (appearing after 12–14 h) were probably caused by the bioluminescence of cells growing in the induction solution, which supports the video record (Appendix A). This accelerated video shows the course of the induction of *P. putida* TVA8 adhered on the OFE cone. A gradual increase in bioluminescence of cells adhering on the APTES modified surface, the base, and the part of the cone, was followed by a high bioluminescence intensity located only on the base. The bioluminescence of these attached cells fell below the detection limits and at the end (12–15 h) a low light signal emerged in the induction solution.

The measured intensities of bioluminescence significantly differed among the five OFEs. Appendix A compares time-records, bioluminescence maxima, and integrals of bioluminescence. OFEs were repeatedly induced over 20 days and bioluminescence always increased after immersion in the toluene induction solution (A 10x, B1 14x, B2 16x, C 15x, D 14x). To use such an OFE biosensor for the detection of toluene, a signal above twice the background noise of the detector (2 × 0.26 nA at 850 V) can reliably confirm the presence of toluene in the liquid sample. This level of bioluminescence generation appeared within 0.5 h after immersion in the induction solution. This growth was observed in all inductions (for all OFEs) with exceptions over the first two days. During this initial period, the cell layers were likely not matured and performed slowly, with low bioluminescent responses.

These results confirmed previous observations that OFEs adhering with *P. putida* TVA8 can be repeatedly used as a detector for toluene after a few days of the stabilization of immobilized cells [19]. A stabilization period of two to four days was observed, even if the cells were immobilized in silica gel [18,28]. Possible ways to improve the stability of bioluminescence signal responses include engineering a cell strain with two reporter genes (one under control of an analyte of interest and one constitutively present to monitor cell viability). An alternative to this is the use of two independent bioreporters (constitutive and inducible) derived from the same strain. Nevertheless, in the optical fiber arrangement, this resolution requires two fibers, which complicates the sensor construction. The analyte-specific signal must be than corrected according to cell viability [29]; or genetical manipulation of a bacterial strains ability to create and dissolve biofilm structure [30,31].

### 3.2. Influence of the Shape of OFE

The calculated characteristics of OFEs are presented in Table 2. Transmittances were calculated numerically as a percentage of rays that pass through the OFE from the wider end to a detector, which is placed on the thin end of the OFE. The number of cells on the wider end of each OFE was determined based on the assumption that the cells are spheres, with diameters of 1 µm, organized in one layer [17]. A product of transmittance and number of cells, referred to as OFE efficiency, is the relative amount of light transmitted by the OFE from a monolayer of cells on the wider end to the detector connected to the OFE thin end [32].

Using the same mathematical model Kalabova et al. [17], we predicted an outcome of experiments with an OFE and a plastic-clad-silica (PCS) fiber, in this study transmittances calculated among the five different OFEs differed by 3.6% at most (see Table 2). These transmittances are negligible in comparison to the changes in bioluminescence production, likely due to a fluctuation in the number of immobilized cells and their physiological state. Intensities of measured bioluminescence (Figure 5) were independent of OFE transmittances and OFE efficiencies. Intensity of bioluminescence production is susceptible to small variations of temperature, pH, medium composition, bioavailability of inducer to each single cell, and many other factors that cannot be completely controlled and increase the higher variability of the bioluminescence. Calculated transmittances of OFEs significantly increased as their shape moved closer to a frustum cone (Appendix A). In reality, OFEs always exhibit such a curved shape.

### 3.3. Immobilization and Induction of Bioluminescence from E. coli 652T7

In the LB_Kan_ growth medium, *E. coli* 652T7 did not adhere on the APTES modified base of the OFEs regardless of the addition of ferric chloride which was added to theoretically enhance the adhesion of microorganisms by lowering the repulsion forces [26]. Cell attachment was observed only at the interface of growth medium and air (Figure 8). At this interface, photon-OFE binding efficiency is <1%, thus bioluminescence of these attached cells did not significantly contribute to detected light.

*E. coli* 652T7 was immobilized on the base of OFE-D with PEI. The time records of daily inductions with LB medium is presented in Figure 9. Other than the first induction intensities, bioluminescence increased within 15 min after immersing the OFE into the LB solution. The intensities remained stable for 18 h on the first day, 6–9 h on all other days, and then sharply decreased due to a depletion of nutrients. To test the OFE with immobilized *E. coli* 652T7 as a biosensor for biotoxicity, HCl was added to the induction solution on the eighth day. This caused pH lowering to pH = 6 and decreased the bioluminescence, which did not recover after the following two inductions. These results imply that an OFE immobilized with *E. coli* 652T7 is sensitive to influences that affect cell viability but cannot be repetitively used as a biosensor since the cells are dying and not recovering.

## 4. Conclusions

In this study, we immobilized the bioluminescent bioreporter *P. putida* TVA8 on a tapered OFE in order to prepare a biosensor for the detection of liquid toluene. This study broadened our previous research, where we used physico-chemical models, using contact angles and zeta potential, to facilitate the attachment of *P. putida* TVA8 to quartz surfaces after treatment with APTES. The biofilm development of *P. putida* TVA8 with time was quantified and the repeatability of the biofilm preparation and the repeatability of bioluminescence detection was determined. Other than a short maturation period (~5 days), the OFEs exhibited a stable bioluminescent response for at least 20 days.

We additionally immobilized the constitutively bioluminescent toxicity bioreporter *E. coli* 652T7 on a PEI modified OFE and demonstrated its potential use as a biosensor for cytotoxicity. Additionally, the immobilization process that we used, without any bulky matrix requirements, could be applied towards many other microbial bioreporters for the biosensing of a variety of different analytes. However, since the reproducibility of the bioreporter responses remains low, the developed biosensor can be used for online, rapid and multiplexed monitoring of the presence of a pollutant, but not its concentration.

## Figures and Tables

**Figure 1 sensors-20-03237-f001:**
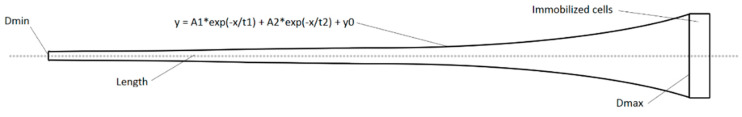
Diagrammatic representation of a tapered optical fiber element (OFE).

**Figure 2 sensors-20-03237-f002:**
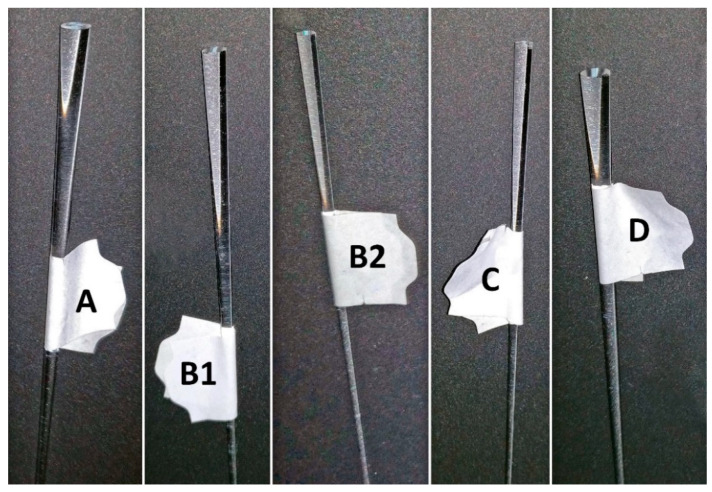
Photographs displaying the wider ends of the OFEs used in this study.

**Figure 3 sensors-20-03237-f003:**
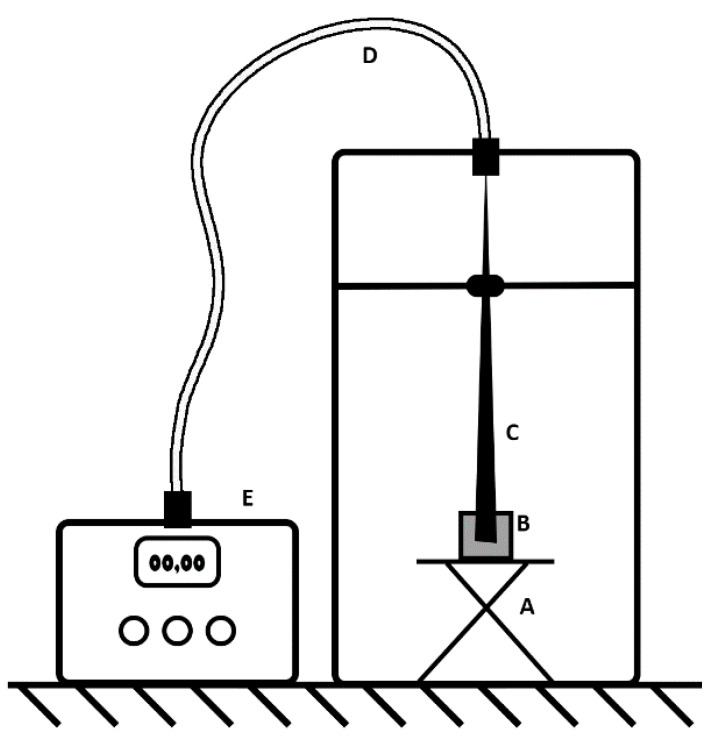
Experimental set-up for monitoring bioluminescence from *P. putida* TVA8 and *E. coli* 652T7 bioreporters adhered to surface modified OFEs. All experiments were performed in a light-tight box that contained an adjustable stand (A), the induction solution (B), the OFE (C), and a light guiding cable (D) that terminated to a photon multiplier measurement device (E).

**Figure 4 sensors-20-03237-f004:**
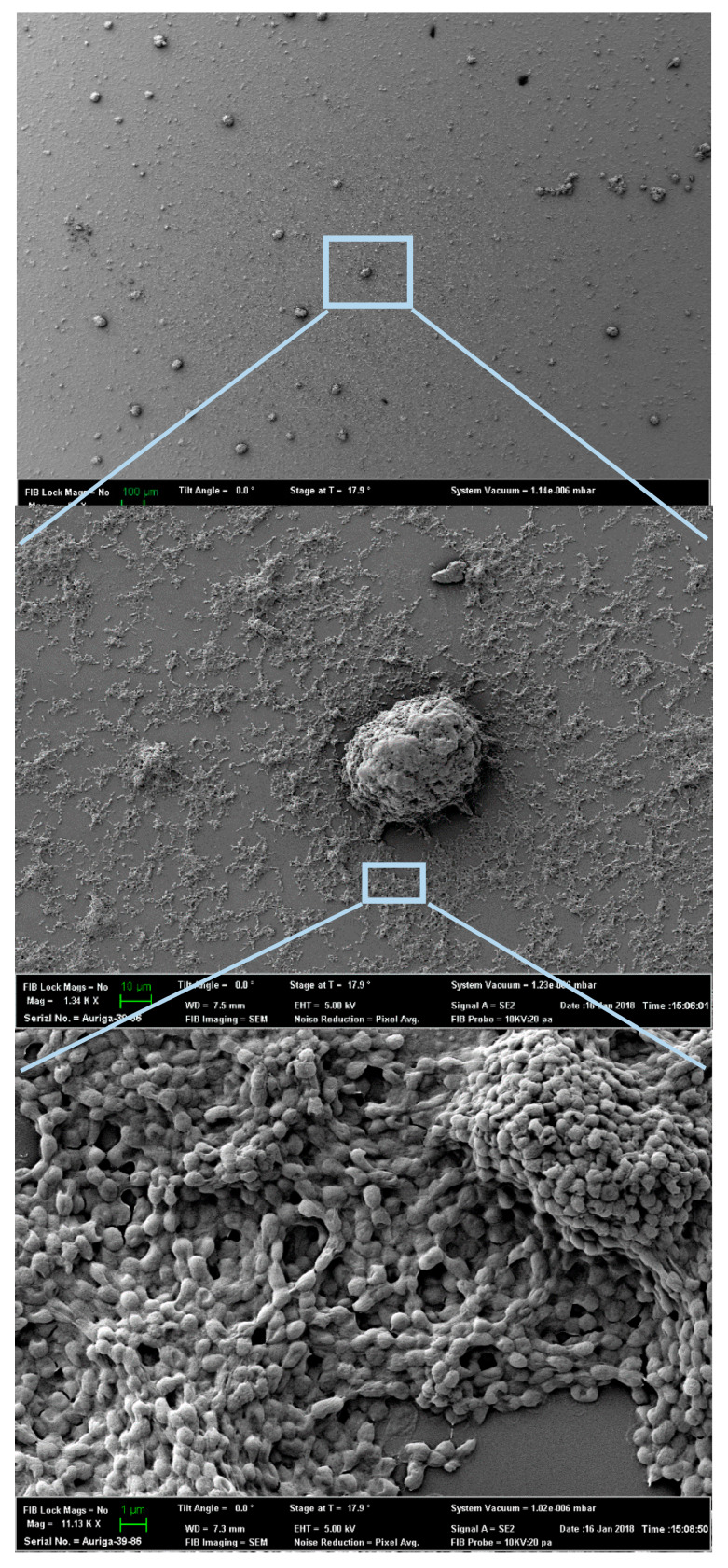
Scanning Electron Microscopy (SEM) image of *P. putida* TVA8 on the (3-aminopropyl) (APTES) modified OFE after the second day of adsorption.

**Figure 5 sensors-20-03237-f005:**
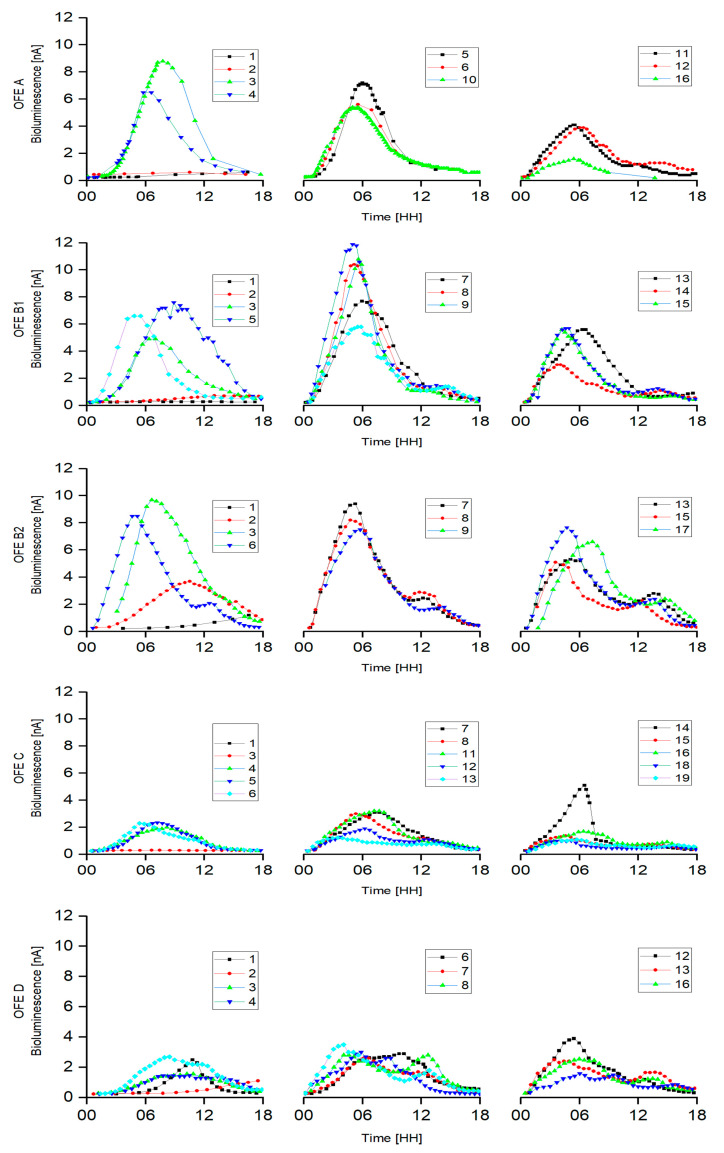
Time-records of daily inductions of bioluminescence of *P. putida* TVA8 immobilized on each of the five available OFEs (A, B1, B2, C, D). Background noise of 26 nA was subtracted from the measured data. *Y*-axis denotes detected bioluminescence intensity in nA. *X*-axis denotes time from the induction of bioluminescence in hours. Each experiment lasted 15–19 days. Chart legends denote the measurement day number.

**Figure 6 sensors-20-03237-f006:**
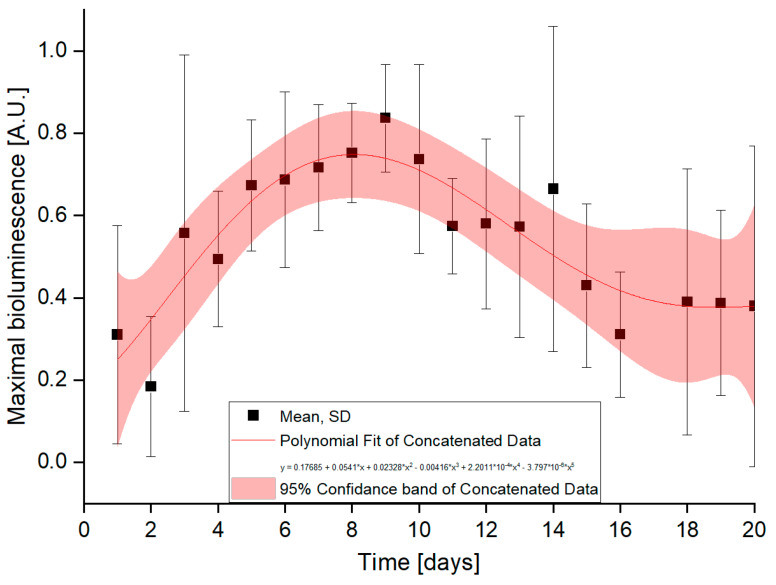
Daily bioluminescence maxima normalized to 1. Aggregated data from the five OFEs plus one OFE from reference [19].

**Figure 7 sensors-20-03237-f007:**
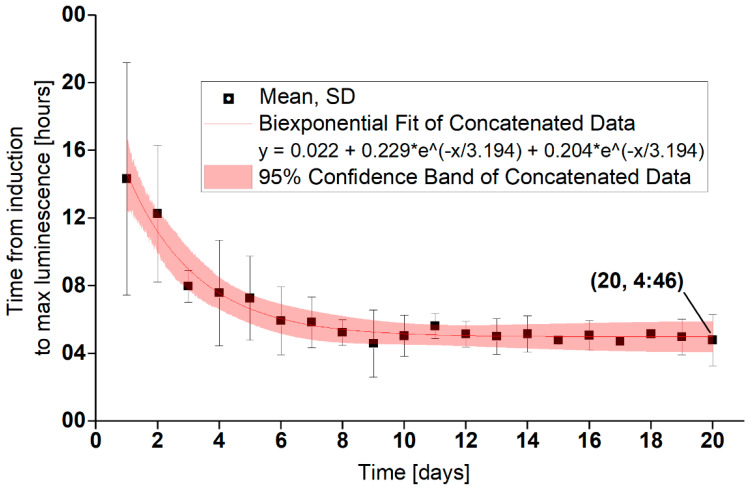
Time of the first bioluminescence maxima. Aggregated data from the five OFEs plus one OFE from reference [19].

**Figure 8 sensors-20-03237-f008:**
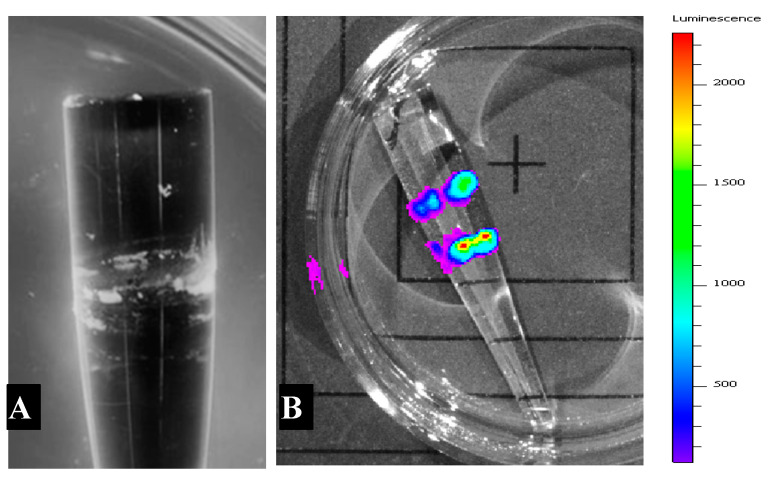
(**A**) *E. coli* 652T7 on the surface of an APTES modified quartz cone after four days in Lauria-Bertani (LB)_Kan_ cultivation medium. (**B**) Bioluminescence of *E. coli* 652T7 induced after immersion in LB medium as measured in an IVIS Lumina K imager (photons/sec/cm^2^/steradian)

**Figure 9 sensors-20-03237-f009:**
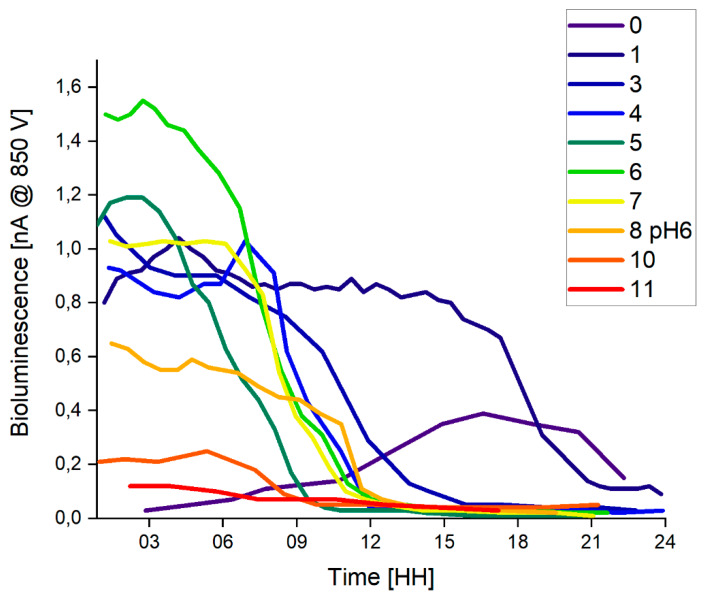
Time records of bioluminescence of *E. coli* 652T7 immobilized on OFE D in polyethyleneimine (PEI). Legend denotes the days after the immobilization.

**Table 1 sensors-20-03237-t001:** Designation of OFEs used in this study. OFEs were approximated using the bi-exponential equation y = y0 + A1*exp(-x/t1) + A2*exp(-x/t2). OFE diameters (*Length*, *Dmax*, *Dmin*) and equation parameters (*A1, t1, A2, t2, y0* = 0) are listed.

OFE	Length [mm]	Dmax [mm]	Dmin [mm]	A1	t1	A2	t2
A	328.5	4.97	0.84	3.093	52.826	1.73	429.369
B1	424	4.1	0.73	3.117	69.541	0.984	1501.727
B2	268	4.1	0.89	3.259	73.314	0.832	12963.443
C	532.5	3.04	0.5	1.789	55.036	1.289	554.939
D	208.8	4.85	1.13	1.606	29.886	3.197	186.254

**Table 2 sensors-20-03237-t002:** Calculated transmittance, cell number and efficiency of the five OFEs used in this study.

OFE	Transmittance [%]	Number of Cells × 10^7^	OFE Efficiency × 10^5^
A	1.41	2.47	3.48
B1	1.62	1.68	2.72
B2	2.21	1.68	3.71
C	1.51	0.92	1.38
D	5.02	2.35	11.75

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
