# Peer review of "Repetitive Detection of Aromatic Hydrocarbon Contaminants with Bioluminescent Bioreporters Attached on Tapered Optical Fiber Elements"

_sensors, 2020, doi:10.3390/s20113237_

Round 1

Reviewer 1 Report

Article “Repetitive detection of aromatic hydrocarbon contaminants with bioluminescent bioreporters attached on tapered optical fiber elements” is devoted to investigation of the repetitive detection of toluene on a tapered optical fiber element (OFE) with an attached layer of Pseudomonas putida TVA8 bioluminescent bioreporters. Article is written at a good scientific level and corresponds to the theme of the journal Sensors.

Nonetheless, to improve the article, I recommend that the authors pay attention to the following comments:

  1. I believe that in the introduction it is necessary to briefly describe the problem of detecting toluene in the environment. This will improve understanding of the importance of research.
  2. In paragraph 2.3, it would be nice to describe in more detail the procedure of OPE preparation.
  3. In the discussion of the results, possible ways to improve the reproducibility for the concentration monitoring of toluene should be added.

There are some inaccuracies in the article:

  1. On page 2 line 93 and page 4 line 140 it is necessary to specify the units for measuring the concentration of solutions.
  2. The title of Fig. 5 (line 206) contains a typo (Ttime-records).
  3. In Fig. 5 hours are designated as “HH”, but in Fig. 9 as “hours”. The uniformity in the designation of the units used is necessary.

I believe that after improvement, the article can be published in the journal Sensors.

Author Response

Dear Reviewer,

Thank you for giving us the opportunity to submit the revised manuscript “Repetitive detection of aromatic hydrocarbon contaminants with bioluminescent bioreporters attached on tapered optical fiber elements” for publication in the special issue of Sensors journal named “Fiber Optic Sensors in Chemical and Biological Applications”. We appreciate the time and effort that you dedicated to providing feedback on our manuscript. We are grateful for the insightful comments on and valuable improvements to our paper. We have replied to all points made by the Reviewers and Assistant Editor and incorporated the suggestions to the manuscript. Please excuse the three-day extension of the submission date, the works on updating our manuscript and responding to the reviewer’s comments didn’t go as smoothly due to the current Covid situation and related restrictions. Please see the responses, in red, below the attached Reviewer report form (reports from both Reviewers are included). The changes to manuscript are highlighted in yellow (Reviewer 1) and blue (Reviewer 2).

Point 1: I believe that in the introduction it is necessary to briefly describe the problem of detecting toluene

Response 1: We agree with the Reviewer. Paragraph about the significance of detection of toluene, related chemicals, and potential health risks was added at the beginning of the article introduction.

Point 2: In paragraph 2.3, it would be nice to describe in more detail the procedure of OPE preparation.

Response 2: Thank you for this suggestion. Info about OFE preparation was added to the paragraph. Reference to the detailed procedure and supplement figure S1 with a scheme of OFEs preparation was added.  

Point 3: In the discussion of the results, possible ways to improve the reproducibility for the concentration monitoring of toluene should be added.

Response 3: Thank you for the suggestion. Cellular bioreporters generally have low reproducibility of bioluminescence responses in long term monitoring (weeks to months). This is due to changes of cell density (cell division or cell dying depending on the nutrient availability) as well as the microbial physiology (transition to stationary phase, transition to swarming behavior upon transition to biofilm). In biofilm, also the changes of nutrient and oxygen availability in different layers of the biofilm due to high cellular density must be considered. As suggested by the reviewer, we have added a note about possible ways to improve the reproducibility for the concentration monitoring of toluene in the discussion part of the article (end of the chapter 3.1.) – these include Engineering a cell with two reporter genes, one under the control of an analyte and the other constitutively present to monitor viability; and genetical engineering of a bacterial strains ability to create and dissolve biofilm structure. (references are listed in the article). Another way is to simultaneously use constitutive bioreporter as an indicator of the physiological state and inducible bioreporter for analyte estimation. Nevertheless, in fiber-optic arrangement this requires the use of two fibers which complicates the long-term detection.

Point 4: Inaccuracies in the article.

  1. On page 2 line 93 and page 4 line 140 it is necessary to specify the units for measuring the concentration of solutions.
  2. The title of Fig. 5 (line 206) contains a typo (Ttime-records).
  3. In Fig. 5 hours are designated as “HH”, but in Fig. 9 as “hours”. The uniformity in the designation of the units used is necessary.

Response 4: We apologize for the inaccuracies mentioned above. All of them were corrected.

Reviewer 2 Report

This manuscript describes a repetitive detection of aromatic hydrocarbon contaminants with bioluminescent bioreporters attached on tapered optical fiber elements. Several comments should be resolved before its publication. The details are shown as below:

  1. The authors state that the bioluminescent cell layer was attached on polished quartz through the modified APTES. However, no evidence was given to prove the successful modification of APTES. The authors should at least cite some reference to support the protocol. For example, CHINESE JOURNAL OF CHEMICAL ENGINEERING, 2017, 25(5), 587-594; SENSORS, 2018, 18(12), 4461; et al.
  2. The authors state that lumps of cell colonies (100-1000 μm apart) among much smaller scattered colonies or single cells were observed under SEM. However, from Figure 4, only individual clusters were observed, and we cannot distinguish whether it is cell colonies. The author should give more evidence.
  3. For SEM images of Figures, the real size bar from the testing machine should also applied.
  4. The figure legends in Figure 5 are meaningful and misunderstanding. The authors should explain these numbers.
  5. The specificity of such tapered OFE should be tested by being incubated with the mixed samples of P. putida TVA8, E. coli 652T7 and other bio-samples.

Author Response

Dear Reviewer,

Thank you for giving us the opportunity to submit the revised manuscript “Repetitive detection of aromatic hydrocarbon contaminants with bioluminescent bioreporters attached on tapered optical fiber elements” for publication in the special issue of Sensors journal named “Fiber Optic Sensors in Chemical and Biological Applications”. We appreciate the time and effort that you dedicated to providing feedback on our manuscript. We are grateful for the insightful comments on and valuable improvements to our paper. We have replied to all points made by the Reviewers and Assistant Editor and incorporated the suggestions to the manuscript. Please excuse the three-day extension of the submission date, the works on updating our manuscript and responding to the reviewer’s comments didn’t go as smoothly due to the current Covid situation and related restrictions. Please see the responses, in red, below the attached Reviewer report form (reports from both Reviewers are included). The changes to manuscript are highlighted in yellow (Reviewer 1) and blue (Reviewer 2).

Point 1: The authors state that the bioluminescent cell layer was attached on polished quartz through the modified APTES. However, no evidence was given to prove the successful modification of APTES. The authors should at least cite some reference to support the protocol. For example, CHINESE JOURNAL OF CHEMICAL ENGINEERING, 2017, 25(5), 587-594; SENSORS, 2018, 18(12), 4461; et al.

Response 1: Thank you for the comment and suggestion. As suggested, we have added the reference in the text. Although it is important to note that ideal APTES modification is not necessary in the case of biofilm OFE biosensor as it is in the case of surface plasmon sensors described in the article.

Point 2: The authors state that lumps of cell colonies (100-1000 μm apart) among much smaller scattered colonies or single cells were observed under SEM. However, from Figure 4, only individual clusters were observed, and we cannot distinguish whether it is cell colonies. The author should give more evidence.

Response 2: Thank you for the comment. Additional images were added, providing a better notion about the colonies/clusters and cells surface distribution.

Point 3: For SEM images of Figures, the real size bar from the testing machine should also applied.

Response 3: We do agree with Reviewer. Detailed device legend, including original size bar, was added to the SEM images.

Point 4: The figure legends in Figure 5 are meaningful and misunderstanding. The authors should explain these numbers.

Response 4: Thank you for the comment. As a response to this point, we have extended the Figure 5 description.

Point 5: The specificity of such tapered OFE should be tested by being incubated with the mixed samples of P. putida TVA8, E. coli 652T7 and other bio-samples.

Response 5: Thank you for the comment. Nevertheless, it is impossible to incubate jointly (in one flask)   P. putida TVA8, E. coli 652T7, or other available strains (ig. Saccharomyces cerevisiae BLYR) since each microorganism needs distinct conditions for optimal production of bioluminescence as temperature, composition of induction media etc. In contrast to portable biosensor (Aldo Roda, Luca Cevenini, Elisa Michelini, Bruce R. Branchini: A portable bioluminescence engineered cell-based biosensor for on-site applications,   Biosensor and Bioelectronic 26 (2011) 3647-3653.  dx.doi.org/10.1016/j.bios.2011.02.022), which is based on multiwall cartridge connected to the photon multiplier by optical fiber tapper, OFE was constructed for monitoring in remote localities (via connection of OFE with standard optical fiber, which transmitted light to the distant detector)  and or repeated measurement in a laboratory.

Selectivity  of P. putida TVA8 , free cells and cells immobilized in silica gel, was described by Kuncová et al (2011). [20]

Round 2

Reviewer 2 Report

All the comments are solved well. And acceptance is suggested.